# The Cognitive Improvement and Alleviation of Brain Hypermetabolism Caused by FFAR3 Ablation in Tg2576 Mice Is Persistent under Diet-Induced Obesity

**DOI:** 10.3390/ijms232113591

**Published:** 2022-11-05

**Authors:** Maite Solas, Marta Zamarbide, Carlos G. Ardanaz, María J. Ramírez, Alberto Pérez-Mediavilla

**Affiliations:** 1Department of Pharmacology and Toxicology, University of Navarra, 31008 Pamplona, Spain; 2IdISNA, Navarra Institute for Health Research, 31008 Pamplona, Spain; 3Neuroscience Program, Center for Applied Medical Research (CIMA), University of Navarra, 31008 Pamplona, Spain; 4Department of Biochemistry and Genetics, University of Navarra, 31008 Pamplona, Spain

**Keywords:** FFAR3, cognition, Alzheimer’s disease, ^18^F-FDG-PET, metabolism, glucose homeostasis, obesity, diabetes, aging

## Abstract

Obesity and aging are becoming increasingly prevalent across the globe. It has been established that aging is the major risk factor for Alzheimer’s disease (AD), and it is becoming increasingly evident that obesity and the associated insulin resistance are also notably relevant risk factors. The biological plausibility of the link between high adiposity, insulin resistance, and dementia is central for understanding AD etiology, and to form bases for prevention efforts to decrease the disease burden. Several studies have demonstrated a strong association between short chain fatty acid receptor FFAR3 and insulin sensitivity. Interestingly, it has been recently established that FFAR3 mRNA levels are increased in early stages of the AD pathology, indicating that FFAR3 could play a key role in AD onset and progression. Indeed, in the present study we demonstrate that the ablation of the *Ffar3* gene in Tg2576 mice prevents the development of cognitive deficiencies in advanced stages of the disease. Notably, this cognitive improvement is also maintained upon a severe metabolic challenge such as the exposure to high-fat diet (HFD) feeding. Moreover, FFAR3 deletion restores the brain hypermetabolism displayed by Tg2576 mice. Collectively, these data postulate FFAR3 as a potential novel target for AD.

## 1. Introduction

Alzheimer’s disease (AD) is a neurodegenerative disease positioned among the most urgent health issues at a global level. This status is due to AD constituting around 60–70% of all cases of dementia, as well as AD-related deaths having increased by 145.2% from year 2000 to year 2019 [1]. These numbers are expected to increase, as AD is an age-related disease, and life expectancy is constantly expanding globally. Crucially, the vast majority (approximately 90–95%) of AD patients suffer from sporadic AD, featuring a multi-factorial pathogenesis that results from an interplay of genetic and environmental risk factors [2]. Among these, states of systemic metabolic dysregulation such as obesity and type 2 diabetes mellitus (T2DM) stand as especially remarkable [3].

A growing body of evidence indicates that T2DM is strongly associated with cognitive dysfunction and memory decline, and the eventual development of AD-type dementia [4,5,6,7,8]. This phenomenon is not surprising, provided that subsets of AD patients exhibit certain T2DM features, such as high fasting insulin levels, insulin resistance (i.e., the resistance of tissues in charge of glucose disposal to the actions of insulin), and glucose intolerance [9]. The causal directionality of this relation is believed to be determined by T2DM-associated systemic insulin resistance being one of the most important risk factors for AD [10,11]. Mechanistically, defective brain insulin signaling as a consequence of T2DM, especially at the hippocampal level, appears to give rise to cognitive dysfunction and AD [12,13,14]. Notably, one of the main consequences of central insulin resistance is a substantial lowering of brain glucose availability [15,16,17]. Intriguingly, a lowering of brain glucose availability is precisely present in prospective AD patients years before the onset of the clinical symptoms, and its detection by neuroimaging studies is sufficient to robustly predict a future AD diagnosis [18,19,20]. Nonetheless, during the course of AD, a glucose hypermetabolic phenotype is acquired by several regions within the AD-affected brains, revealing complex dynamics of glucose metabolism in different AD phenotypes and AD-affected brain regions [21,22,23]. The identification of these puzzling phenomena further highlighted the relevance of disrupted glucose metabolism across the whole AD pathogenic process.

At the same time, obesity is one of the main causes of systemic insulin resistance via various adipose tissue-dependent mechanisms. This phenomenon potentially leads to pancreatic β cell dysfunction and the subsequent failure to control glycemia, resulting in T2DM [24]. Several prospective epidemiological studies have also identified that obesity increases the risk of developing AD later in life [25,26,27], evidencing a continuum of obesity, T2DM, and AD [28]. Deciphering the mechanisms underlying this intricate relationship may reveal unique insights on the pathogenesis of AD, and shed light on how to develop optimized strategies for the prevention and treatment of AD.

Free fatty acids (FFAs) contribute to a plethora of disturbances regarding carbohydrate metabolism, among which insulin resistance, glucose intolerance, and disrupted pyruvate oxidation have been clearly identified [29]. In fact, fasting plasma levels of FFAs are elevated in obesity [30,31,32,33] and are sufficient to predict the incidence of T2DM [34,35,36,37,38,39,40]. Establishing a causal directionality, the hyperinsulinemia and insulin resistance leading to T2DM in obese individuals are believed to be a response to elevated FFAs in these subjects [41]. FFAs can be classified regarding their carbon chain length, resulting in subdivisions termed short-chain fatty acids (SCFAs), which feature 1–6 carbon atoms, medium-chain fatty acids (MCFAs), which incorporate 7–12 carbon atoms, and long-chain fatty acids (LCFAs), with more than 12 carbon atoms [42]. SCFAs are especially interesting for the prevention of obesity and T2DM, as dietary fiber protects from these metabolic disturbances by undergoing bacterial fermentation in the large intestine, producing SCFAs that mediate reductions in body weight and appetite, and induce benefits in glucose tolerance [43]. Thus, SCFAs protect obese mice from excessive body weight and glucose intolerance via mechanisms that are at least partly mediated by two out of the five FFA receptors (FFARs), namely FFAR2 and FFAR3 [43,44]. FFAR2 and FFAR3 present similar expression patterns throughout the organism, being present in pancreatic islets as well as in enteroendocrine cells of the intestine, but some differences should be noted: only FFAR2 is present in adipocytes and certain immune cells, while only FFAR3 is expressed by sympathetic ganglion cells and enteric neurons [42].

An accurate clarification of the differential contribution of FFAR2 and FFAR3 to metabolic benefits and disturbances has remained elusive. Furthermore, different studies performed in either FFAR2 or FFAR3 null mice have led to conflicting results describing both increased and reduced body weight and glucose tolerance upon FFAR2 or FFAR3 loss [45,46,47,48,49]. Nonetheless, regarding the nervous system, FFAR3 has been confirmed as the mediator of the effects exerted by SCFAs on sympathetic nervous system activity, therefore participating in the control of systemic energy homeostasis [50]. Crucially, the hippocampal expression of both FFAR2 and FFAR3 was recently identified, and a striking 5-fold upregulation of FFAR3, but not FFAR2, expression was identified in initial stages of AD [51]. Additionally, the same study demonstrated that the loss of FFAR3 is sufficient to reverse the cognitive impairment and amyloid-β (Aβ) pathology exhibited by the Tg2576 AD mouse model, which overexpresses the amyloid precursor protein (APP) carrying the Swedish mutation [51].

In light of these compelling findings, several questions arise regarding the mechanism of FFAR3 loss-elicited cognitive improvement, its potential therapeutic application and, more generally, the specific role of FFAR3 for metabolism and cognition. Thus, considering the importance of FFAs for obesity and T2DM, and the postulated continuum of obesity, T2DM, and AD: (i) is the reversal of AD cognitive dysfunction upon FFAR3 inactivation due to an improvement in energy homeostasis induced by FFAR3 ablation? Additionally, if such scenario exists, (ii) would the observed cognitive improvement resist upon a metabolic challenge, such as high-fat diet (HFD) feeding? To address these unknowns, in the present study, FFAR3 null (FFAR3^−/−^) mice were crossed with the Tg2576 AD mouse model, obtaining Tg-FFAR3^−/−^ mice. These mice were exposed to a metabolic challenge consisting of a 6-month-long HFD feeding from the 9th to the 15th month of life, and subjected to a metabolic and cognitive examination.

## 2. Results

### 2.1. FFAR3 Ablation-Induced Systemic Glucose Homeostasis Impairment Is Reverted in Tg2576 Mice

In an attempt to determine whether FFAR3 is involved in the maintenance of systemic energy homeostasis, we investigated if mice lacking FFAR3 display alterations in glucose handling and insulin sensitivity.

We first analyzed systemic energy homeostasis in basal conditions, i.e., when mice are fed with a normal chow diet (NCD) (Figure 1a). A body weight assessment indicated that, although Tg2576 mice feature a significantly lower body weight (two-way ANOVA, main effect of Tg, F_1,27_ = 10.83, *p* < 0.01), FFAR3 ablation induces a significant body weight increase (two-way ANOVA, main effect of FFAR3, F_1,27_ = 14.98, *p* < 0.001) (Figure 1b). In line with a lower body weight, Tg2576 mice showed improved glucose tolerance (for AUC, two-way ANOVA, main effect of Tg, F_1,27_ = 8.302, *p* < 0.01), as they demonstrated a higher ability to readjust systemic glucose levels after hyperglycemia induced by peripheral glucose administration (Figure 1c,d). This improved glucose tolerance was not accompanied with an enhanced insulin sensitivity, as all the groups suffered from a similar degree of hypoglycemia upon an exogenous administration of insulin (Figure 1e,f).

To study if this improved glucose tolerance showed by Tg2576 mice is maintained upon a metabolic challenge, we exposed 9-month-old mice to a 60% fat-containing diet for 6 months (until 15 months of age) (Figure 2a). Interestingly, FFAR3^−/−^ mice showed a significantly higher body weight when subjected to this HFD, an attribute that was completely reversed when FFAR3 ablation was induced in Tg2576 mice (two-way ANOVA, significant interaction, F_1,27_ = 5.735, *p* < 0.05, followed by Tukey ** *p* < 0.01 and *** *p* < 0.001) (Figure 2b). This feature evidenced that, strikingly, FFAR3 ablation-facilitated obesity is not present in the Tg2576 AD mouse model. Furthermore, this body weight phenotype is accompanied by a marked glucose intolerance in the FFAR3^−/−^ group, which is again restored in Tg-FFAR3^−/−^ mice (for AUC, two-way ANOVA, significant interaction, F_1,26_ = 4.394, *p* < 0.05, followed by Tukey * *p* < 0.05, ** *p* < 0.01 and *** *p* < 0.001) (Figure 2c,d). In a similar manner to the phenotype observed in NCD-fed mice, the reported improvement in glucose tolerance was not due to enhanced insulin sensitivity (Figure 2e,f). Together, these data demonstrate that the need for FFAR3 to prevent HFD-induced obesity and glucose intolerance does not operate in Tg2576 mice, revealing that Tg-FFAR3^−/−^ mice indeed feature enhanced metabolic resilience.

### 2.2. FFAR3 Ablation Decreases the Brain Glucose Hypermetabolism Observed in Tg2576 Mice

Age-related brain energy alterations have been related to neurodegenerative diseases, with a brain hypermetabolic phenotype being present in several brain regions in confirmed AD patients [21,22]. Indeed, the Tg2576 AD mouse model shows brain glucose hypermetabolism as assessed with ^18^F-FDG-PET [52]. Congruently, in our hands, Tg2576 mice showed a significantly increased brain glucose metabolism, a phenomenon that was completely abrogated upon FFAR3 ablation (two-way ANOVA, significant interaction, F_1,16_ = 8.369, *p* < 0.05, followed by Tukey * *p* < 0.05) (Figure 3a,b). Notably, the brain glucose hypermetabolism observed in Tg2576 mice was further exacerbated by long-term HFD feeding and, again, this phenomenon was reversed to control-like levels in the Tg-FFAR3^−/−^ group (two-way ANOVA, significant interaction, F_1,14_ = 17.74, *p* < 0.001, followed by Tukey * *p* < 0.05 and ** *p* < 0.01) (Figure 3c,d).

### 2.3. FFAR3 Ablation Restores the Cognitive Impairment of Tg2576 Mice

A recent study has already demonstrated that genetic inactivation of FFAR3 reverses the cognitive impairment showed by Tg2576 mice [51]. However, that study tested the mice at the age of 12 months, i.e., the age when the characteristic cognitive impairment present in Tg2576 begins to be manifested. Therefore, in the present study we decided to test the animals at a more advanced age (15 months) to examine whether the cognitive improvements induced by FFAR3 ablation are maintained at a more drastic aging condition. To this end, spatial reference learning and memory function were assessed using the Morris water maze (MMW) paradigm. After excluding visuomotor deficits in the habituation phase (data not shown), we observed that the FFAR3 ablation-induced cognitive improvement was still preserved in 15-month-old Tg-FFAR3^−/−^ mice not only in the memory acquisition phase (repeated measurements ANOVA, F_3,243_ = 3.578, *p* < 0.05) (Figure 4a) but also in the memory retention phase (third probe trial, two-way ANOVA, significant interaction, F_1,25_ = 4.556, *p* < 0.05, followed by Tukey * *p* < 0.05 and ** *p* < 0.01) (Figure 4b). Therefore, it can be concluded that the FFAR3 ablation-elicited the prevention of cognitive dysfunction in the context of AD is persistent until advanced stages of the disease in mice.

Intriguingly, the close relationship between HFD feeding and dementia has been extensively described, and obesity, T2DM, and AD seem to be intricately interconnected (see Section 1). Therefore, we aimed to assess whether the improved cognitive performance of Tg-FFAR3^−/−^ mice was preserved when subjected to such a metabolic challenge as a long-term HFD feeding. Notably, the cognitive improvement obtained by the FFAR3 deletion in Tg2576 mice was also maintained when these mice were exposed to a HFD, both regarding the memory acquisition (repeated measurements ANOVA, F_3,222_ = 3.096, *p* < 0.05) (Figure 4c) and memory retention phases (third probe trial, two-way ANOVA, significant interaction, F_1,26_ = 5.377, *p* < 0.05, followed by Tukey * *p* < 0.05) (Figure 4d). These results evidence that FFAR3 loss is still sufficient to prevent AD-related cognitive dysfunction even in the scenario of a dietary metabolic challenge, again reinforcing the view of an enhanced metabolic resilience of Tg-FFAR3^−/−^ mice.

## 3. Discussion

In recent years, SCFAs have been identified as key regulators of obesity and T2DM, mediating alterations in adiposity, body weight, and glucose tolerance [43]. At the same time, both obesity and T2DM maintain an inextricable connection with diverse forms of brain illness and neurodegenerative processes [53,54,55]. This series of relationships has led to the suggestion that SCFAs might play hitherto unacknowledged roles in a wide range of brain disorders. In fact, several studies have highlighted the importance of SCFAs for brain function and dysfunction in the context of neuropsychological disorders, autism, energy metabolism, and AD [49,56,57,58]. Focusing on AD, alterations in both SCFA concentrations and SCFA-producing microbiota have been documented in AD patients [59,60,61]. Most strikingly, the existence of a correlation between blood SCFA concentrations and the severity of the AD burden has been recently demonstrated [62].

Besides serving as energy-yielding substrates, SCFAs are also signaling molecules, acting on their endogenous G protein-coupled receptors, especially FFAR2 and FFAR3. Specifically, FFAR3 has been identified as an important member of the SCFA receptor class with substantial implications for health and disease [63]. Due to the elevated expression of FFAR3 in endocrine tissues, including pancreatic islets and intestines, this receptor has been postulated as a key therapeutic target in metabolic diseases. Indeed, considering the close link between metabolic disturbances and dementia, a deep characterization of the role of FFAR3 is extremely pertinent in order to assess its potential role not only in obesity/T2DM, but also in other related diseases, such as AD.

Studies performed in FFAR3 whole-body knockout mice have led to conflicting data regarding the role of FFAR3 in controlling energy balance. While some studies describe that FFAR3^−/−^ mice show decreased adiposity linked to a reduced caloric and SCFA absorption [48], others have demonstrated a reduced energy expenditure and increased adiposity upon FFAR3 deletion [50,64]. In our hands, FFAR3^−/−^ mice showed increased body weight, both when fed with NCD or HFD. This rise in body weight can be directly linked to the hyperphagia that has been previously described in FFAR3^−/−^ mice [49], together with a decreased energy expenditure, as it has been demonstrated that FFAR3 activation promotes sympathetic outflow and contributes to the control of energy expenditure and metabolic homeostasis [50].

Contradictory data has also been yielded concerning the role of FFAR3 on glucose homeostasis. While compelling evidence supports that FFAR3 ablation reduces insulin secretion and hence induces a marked glucose intolerance [64,65,66], others have found that FFAR3^−/−^ mice exhibit improved glucose homeostasis [63,67,68]. These discrepancies could be attributable to the different genetic approaches used to generate the FFAR3^−/−^ mouse model and hence the unique downstream genetic changes present in each model. Another reason could be the variability of the experimental conditions employed for the glucose tolerance assessment, as well as the differences on the mouse model backgrounds. Specifically, in our hands, in parallel to the increased body weight, FFAR3 ablation induced a marked glucose intolerance upon HFD feeding. It has been previously described that FFAR3 in enteroendocrine cells promotes the secretion of peptide YY (PYY), inhibiting food intake [69]. Interestingly, in enteroendocrine L cells, PYY is co-expressed with glucagon-like peptide 1 (GLP1), suggesting that FFAR3 stimulation could also lead to an increase in GLP1 secretion, which also inhibits food intake and stimulates insulin secretion from pancreatic β cells [70]. Consistently, it has been reported that FFAR3^−/−^ mice present reduced GLP1 secretion, not only in vivo, but also in primary colonic cultures [65], suggesting that diminished GLP1 levels could be underlying the impaired glucose homeostasis observed in our FFAR3^−/−^ mice. However, this may not be the only cause underlying this glucose intolerance, as FFAR3 is not exclusively expressed in intestinal cells. Indeed, FFAR3 has also been characterized as playing a role in the effect of SCFA on immune cells and adipocytes [71,72]. Focusing on adipocytes, SCFA have been involved in adipogenesis [73], lipolysis [73], and in the enhancement of leptin production through FFAR3 [74]. Therefore, whole-body ablation of FFAR3 could alter adipocyte function, leptin release, pancreatic β cell function, and GLP1 secretion, impacting glucose tolerance.

Metabolic alterations are an important feature within the AD pathology [10,75], particularly the brain insulin resistance that has been closely tied to cognitive deficiencies [4,13,76]. Extensive literature points towards a link between T2DM, obesity, insulin resistance, and an elevated risk of developing AD and related disorders [77,78]. However, to date, the available data on brain insulin resistance and peripheral metabolic homeostasis in transgenic AD mouse models are limited. Although extensive discrepancies exist between different mouse models, most of the studies agree that Tg2576 mice exhibit increased body weight, fasting hyperglycemia, and hyperinsulinemia. These changes are accompanied by peripheral insulin resistance which coincides with the onset of Aβ accumulation in the brain [79,80,81]. In contrast to these reports, in our hands, Tg2576 mice exhibited intact glucose tolerance and insulin sensitivity, not only on NCD, but also when subjected to HFD. More strikingly, the increased body weight and glucose intolerance developed by FFAR3 ablation under HFD feeding is completely abrogated in Tg-FFAR3^−/−^ mice. These surprising results indicate that, while FFAR3 seems to be essential for the maintenance of a proper energy balance and glucose homeostasis in physiological conditions, this essentiality is completely lost under AD pathological conditions.

Specific and progressive brain energy metabolism reduction is among the most core manifestations of the AD pathology. Indeed, cerebral glucose changes at the hippocampal and entorhinal cortical levels can differentiate individuals that convert from mild cognitive impairment (MCI) to AD from those who remain MCI [82]. In theory, small animal PET imaging in AD transgenic mouse models should be a useful tool to evaluate the metabolic changes observed through the AD pathology. However, cerebral glucose hypometabolism has been shown to be less evident in AD mouse models than in AD patients. Notably, due to the heterogenic phenotypes of the selected transgenic mouse models and methodological differences between laboratories, unchanged [83], elevated [52,84,85,86,87], or diminished [88,89,90,91,92,93] brain glucose metabolism has been described in AD mice. Our results showed a marked brain hypermetabolism in Tg2576 mice both in NCD and HFD. A potential explanation of the observed hypermetabolism is the increased Aβ-related seizure activity demonstrated in AD mouse models [94]. It has been shown via in vivo electrophysiology that amyloidogenic mouse models exhibit increased excitability and nonconvulsive seizure activity [94]. Besides, it has been proposed that neuroinflammation could be another factor behind the increased glucose uptake observed in Tg2576 mice, as activated glial cells also import and metabolize ^18^F-FDG [95,96]. Moreover, other factors could be influencing the increased brain metabolism seen in Tg2576 mice, such as an increased basal metabolic rate, reflecting increased APP expression, and possibly elevated amyloid burden. In this line, several studies have shown that Aβ assemblies create membrane pores [97]. As Aβ can be localized in the mitochondria [98], the formation of Aβ-induced pores in the mitochondrial membrane could deplete the normal ionic gradients established by the sodium pump and other transporters, causing an elevated energetic demand for ionic pumping. Taking into account that the proper maintenance of ion gradients accounts for around 20% of the basal energy needs of the body and that 60% of brain ATP is used for this function, the generation of Aβ-induced pores may increase energy demands [99], leading to the increased basal metabolic rates and hypermetabolic PET scans observed in the present study.

Notably, a previous study addressing the role of FFAR3 in AD showed that Tg-FFAR3^−/−^ mice showed a significant decrease in amyloid plaque burden [51]. As described in that study, cortical homogenates obtained from 12-month-old Tg-FFA3R^−/−^ mice showed barely detectable levels of Aβ_42_ measured by ELISA, as well as a completely absence of Aβ deposits assessed by immunohistochemical analysis in hippocampal and frontal cortex sections of Tg-FFA3R^−/−^ mice. However, APP amyloidogenic processing was not decreased, as no significant differences in the levels of the APP-derived carboxy-terminal fragments (C83 and C99) between Tg-FFA3R^−/−^ and Tg2576 mice were observed. This indicates that the absence of senile plaques in Tg-FFAR3^−/−^ mice could be due to an increase in Aβ clearance and not a decrease in Aβ production. Indeed, authors showed a significant increase in the expression levels of the insulin-degrading enzyme (IDE) in Tg-FFAR3^−/−^ mice that could promote an increase in Aβ clearance that reduces the accumulation of this harmful peptide in the brain. Moreover, 12-month-old Tg-FFA3R^−/−^ mice displayed significantly reduced levels of phosphorylated Tau, directly linked to a decrease in the activity of GSK3β and Cdk5, Tau phosphorylating kinases. As previously mentioned, one of the most accepted explanations for the brain glucose hypermetabolism observed in Tg2576 mice is the accumulation of Aβ. Therefore, it is tempting to speculate that the complete alleviation of brain hypermetabolism observed in Tg-FFAR3^−/−^ mice could be directly linked to a FFAR3 ablation-induced improved Aβ clearance.

The central nervous system (CNS) is an intricate network composed of a myriad of cell types that differ in a vast variety of properties and functions. Notably, the amount of lipids present in the brain represents a larger proportion of mass compared to most other tissues, only behind the adipose tissue [100]. A remarkable difference between these tissues is that, while lipids are stored as energy deposits in the adipose tissue, CNS lipids are responsible of innumerable functions, such as the formation of large cell membrane extensions for proper neural cell crosstalk. Concerning SCFA receptors, in contrast to the expression observed in gut and peripheral organs, most studies have found a very low expression level of FFAR3 in the brain [44,71,101]. However, it has been recently demonstrated that FFARs are differentially expressed in the hippocampus of control subjects’ and AD patients’ brains [51]. In that study, Zamarbide et al. showed that FFAR3 mRNA levels are increased, while FFAR1 and FFAR2 appeared to be downregulated in early stages of the AD pathology, reasoning that FFAR3 could play a key role in AD onset and progression. In that study, mice were tested at an age corresponding to the onset of cognitive impairment manifestations, i.e., 12 months of age, suggesting that FFAR3 ablation could prevent the disease onset. In the present study we assessed if the cognitive improvement induced by FFAR3 ablation is also observed in a more demanding situation: a more advanced age (15 months) and a metabolic challenge (HFD feeding). Notably, FFAR3 ablation was able to restore cognitive performance in 15-month-old Tg2576 mice, not only on NCD but also when challenged with HFD, reinforcing the idea that FFAR3 loss is sufficient to prevent AD-related cognitive dysfunction, even in the scenario of advanced aging and dietary metabolic challenge. The cognitive improvement could be linked to an improved brain connectivity. In this line, Zamarbide et al. showed that the lack of FFAR3 restores the aberrant expression of immediate early genes, such as pCREB, Arc, and c-Fos, in Tg2576 mice making them similar to the levels found in WT mice. Additionally, levels of cytoskeletal proteins of dendritic spines, such as MAP2 and PSD95, were restored in Tg2576 when FFAR3 expression was ablated. Overall, these results reveal that the abnormalities in brain glucose metabolism exhibited by the Tg2576 AD mouse model are corrected by FFAR3 ablation, offering a potential explanation for the improved cognition of Tg-FFAR3^−/−^ mice.

Taken together, these results show that FFAR3 whole-body ablation induces peripheral glucose homeostasis impartment, but Tg-FFAR3^−/−^ mice appear to be resilient to this metabolic alteration. Interestingly, the FFAR3 deletion in Tg2576 mice improves cognitive performance and restores brain hypermetabolism, indicating a potential decrease in the amyloid burden that could offer a plausible explanation for the observed ameliorated cognition. Notably, the cognitive ability preservation occurs not only in the onset of the disease as previously reported, but also in the context of advanced age and dietary metabolic challenge. In the light of a scenario where aging, obesity, and T2DM appear to be closely linked to neurodegeneration, these data postulate FFAR3 as a potential novel target for AD.

## 4. Materials and Methods

### 4.1. Animals

In the present study, the Tg2576 AD mouse model was employed. These mice express the human 695-aa isoform of APP with the double Swedish mutation (APPswe) [(APP695)Lys670 → Asn, Met671 → Leu] driven by a prion promoter. This mouse model features an exponential Aβ peptide accumulation between the 7th and the 12th months of life, which is accompanied by memory impairments in the Morris water maze test at the age of 12–15 months [79].

FFAR3^−/−^ mice were kindly donated by Masashi Yanagisawa [48]. FFAR3^−/−^/Tg2576 mice (named Tg-FFAR3^−/−^ mice) were generated in our laboratories by breeding FFAR3^−/−^ mice with Tg2576 mice [51].

Mice were housed in groups of six in positive pressure-ventilated racks at 25 ± 1 °C, using a 12-h light/12-h dark cycle. Mice had *ad libitum* access to filtered and UV-irradiated water and diet at all times, and food was only withdrawn if required for an experiment. In an attempt to induce a metabolic challenge, mice were fed on a high-fat diet (HFD) containing 35% calories from carbohydrates, 20% from protein, and 45% from fat (coming from soybean oil and lard) (Research Diets, #D12451), starting at 9 months of age.

All animal procedures conducted in the University of Navarra were performed in compliance with the European and Spanish regulations (2003/65/EC; 1201/2005) for the care and use of laboratory animals and approved by the Ethical Committee of the University of Navarra (ethical protocol number 018/05).

### 4.2. Genotyping of Mouse Models

Genotyping was performed by PCR using the following custom-designed primer sets [51]: 5′-CAC ACT GCT CGA TCC GGA ACC CTT and 5′-GAG AAC TGT CTG GAA AAC GCT CAC to identify the mutant *Ffar3* allele, 5′-CGA CGC CCA GTG GCT GTG GAC TTA and 5′-GTA CCA CAG TGG ATA GGC CAC GC to detect the WT allele. To identify the mutant APP allele, we used 5′GTT GAG CCT GTT GAT GCC CG and 5′GTT GAG CCT GTT GAT GCC CG, and to detect the WT allele 5’AAG CGG CCA AAG CCT GGA GGG TG with 5′GTG GAT AAC CCC TCC CCC AGC CTA GAC CA.

### 4.3. Metabolic Studies

All metabolic studies were conducted at 15 months of age, i.e., after a 6-month-long exposure to HFD feeding.

#### 4.3.1. Body Weight

Body weight was measured at 15 months of age in all mice groups under *ad libitum*-fed conditions.

#### 4.3.2. Glucose Tolerance Test

The glucose tolerance test (GTT) was performed in mice that previously underwent a fasting period of 6 h. After the fasting period, the baseline glucose concentration in tail vein-derived blood was measured (0 min) using a handheld glucometer (Accu-Chek Aviva Glucometer, Roche, Basel, Switzerland) and blood glucose test strips (#06453970037; Roche, Basel, Switzerland). Afterwards, each mouse received an i.p. injection of glucose (2 g/kg body weight). Subsequently, the blood glucose concentration was measured 15, 30, 60, and 120 min after the glucose injection.

#### 4.3.3. Insulin Tolerance Test

The insulin tolerance test (ITT) was carried out in *ad libitum*-fed mice. After measuring the baseline glucose concentration in tail vein-derived blood (0 min) with a handheld glucometer (Accu-Chek Aviva Glucometer, Roche, Basel, Switzerland), each mouse received an i.p. insulin administration (0.75 U/kg body weight; Actrapid; Novo Nordisk, Bagsværd, Denmark) and blood glucose concentration was subsequently monitored 15, 30, and 60 min after the insulin injection.

### 4.4. Morris Water Maze

The water maze consisted of a circular pool (diameter of 145 cm) filled with water (21–22 °C) and virtually divided into four equal quadrants. The MWM test was divided into three phases: habituation, acquisition, and retention.

The habituation phase (visible platform test) was conducted the day before the acquisition phase. The habituation consisted of six trials, meaning that each mouse performed the test six times. Each time, the aim of the mouse was to reach the platform, which contains a visual object on top of it, thus ensuring the possibility of visualizing the platform for mice undergoing the test. Each trial finished when the mouse reached the platform (escape latency) or after 60 s, whichever came first. Mice failing to reach the platform were guided onto it and placed on the platform for 15 s.

The acquisition phase was conducted during nine consecutive days. During these days, learning capacity was analyzed. For this purpose, a hidden transparent platform was placed 1 cm below the water surface. Over the nine days, four trials/day were performed, and mice were introduced to the pool from a different quadrant in each of these four daily trials. As in the habituation phase, each trial finished when the mouse reached the platform or after 60 s.

Probe trials (retention phase) were performed at the beginning of the 4th, 7th, and 10th days of the test, with a single trial/day. The platform was removed from the pool and mice were allowed to swim for 60 s, recording the time spent in the target quadrant.

All trials were recorded using a video camera above the center of the pool and connected to a video tracking system (EthoVision XT, Noldus Information Technology, Wageningen, The Netherlands).

### 4.5. ^18^F-FDG-PET

#### 4.5.1. PET Acquisition

Mice were fasted overnight. The day of the study, ^18^F-fluorodeoxyglucose (^18^F-FDG) (9.5 MBq ± 0.6 in 80–100 μL) was administered through the tail vein and placed back in the cage for 50 min. Then, animals were anesthetized with 2% isoflurane and placed in a small animal PET tomograph (Mosaic, Philips, Cleveland, OH, USA) to subsequently acquire the PET signal for 15 min.

#### 4.5.2. PET Analysis

For the semi-quantitative analysis of the in vivo PET images, data was analyzed using the PMOD software (PMOD v3.2, PMOD Technologies Ltd., Adliswil, Switzerland). Images were expressed in standardized uptake value (SUV) units, using the formula SUV = [tissue activity concentration (Bq/cm^3^)/injected dose (Bq)] × body weight (g).

### 4.6. Statistical Analysis

Results, reported as means ± SEM, were analyzed by Prism 9.0 (GraphPad Software, San Diego, CA, USA). Normality was checked by the Shapiro–Wilk’s test (*p* < 0.05). Data containing two variables (four experimental groups) were analyzed with two-way ANOVA, followed by Tukey. In GTT, ITT, and MWM acquisition test over-all treatment effects were examined by two-way repeated measures ANOVA. In all cases, the significance level was set at *p* < 0.05.

## Figures and Tables

**Figure 1 ijms-23-13591-f001:**
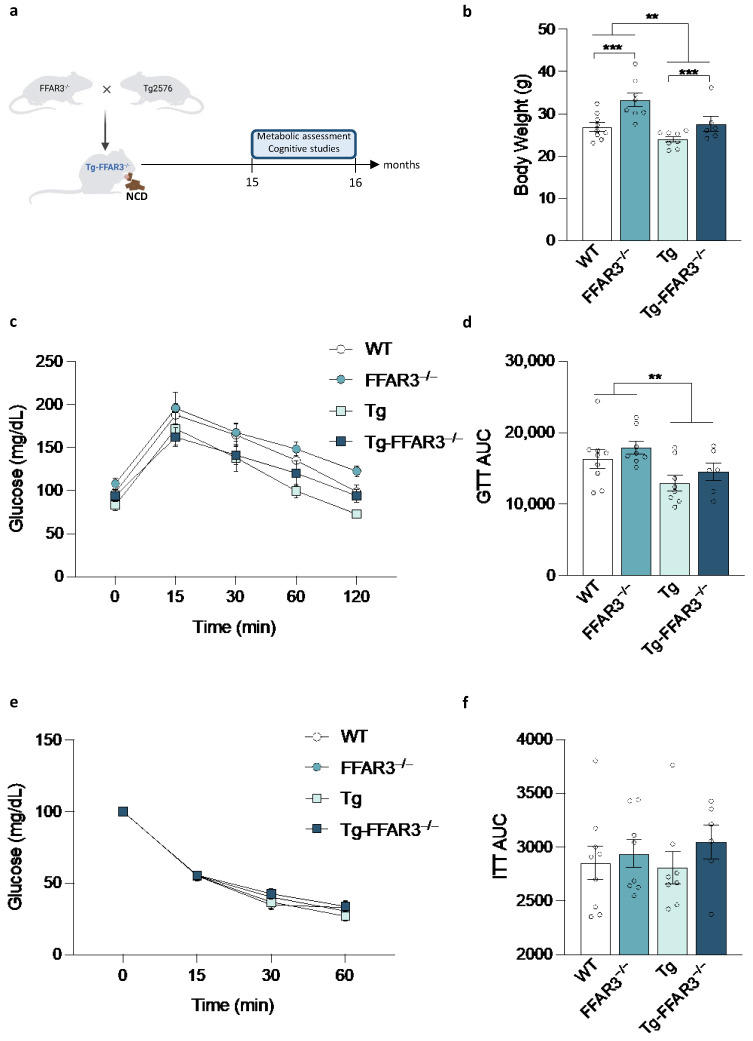
FFAR3 ablation induces body weight elevation in mice fed with normal chow diet (NCD). (**a**) Schematic representation of the strategy used to generate Tg2576 mice lacking FFAR3 and the subsequent metabolic assessment. (**b**) Body weight evaluation of wild type (WT), FFAR3 knockout (FFAR3^−/−^), Tg2576 (Tg) and Tg2576 FFAR3 knockout mice (Tg-FFAR3^−/−^). (**c**) Glucose tolerance test (GTT) and (**d**) the corresponding AUC quantification. (**e**) Insulin tolerance test (ITT) and (**f**) the corresponding AUC quantification. Results are expressed as mean ± SEM and circles in the subfigures represent each individual value (*n* = 6–9 in each group). ** *p* < 0.01, *** *p* < 0.001, two-way ANOVA followed by Tukey.

**Figure 2 ijms-23-13591-f002:**
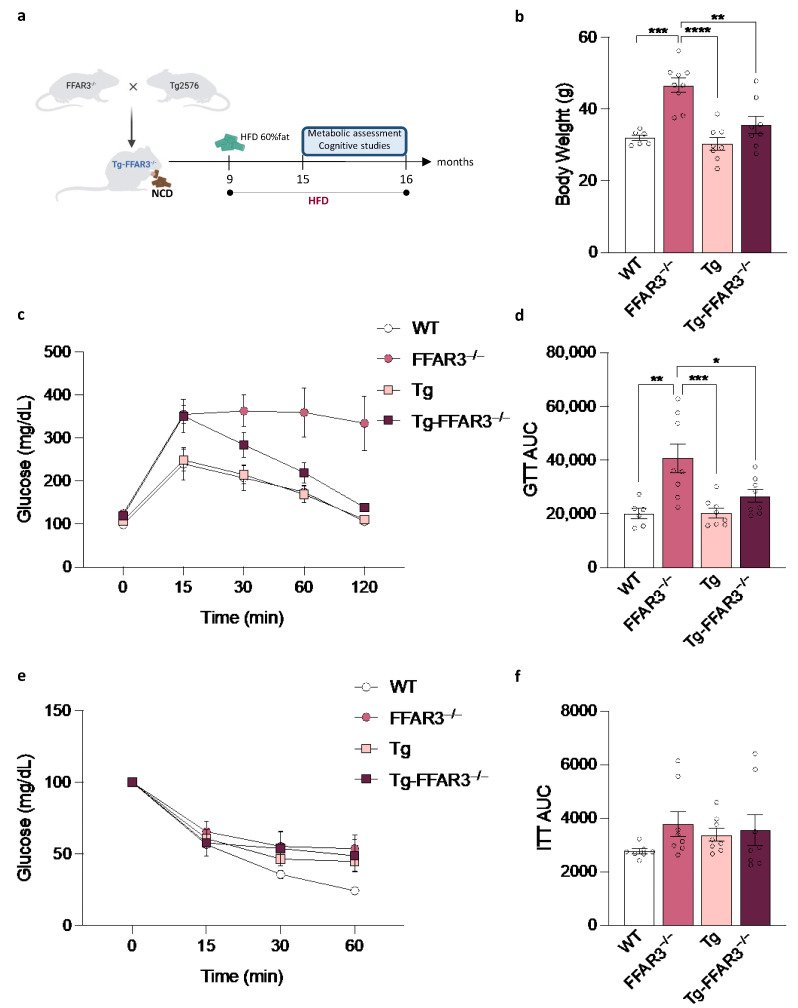
FFAR3 ablation-induced body weight elevation and glucose intolerance is not present in Tg2576 mice fed with high-fat diet (HFD). (**a**) Schematic representation of the strategy used to generate Tg2576 mice lacking FFAR3, dietary intervention with HFD (starting at 9 months of age), and the subsequent metabolic assessment (at 15 months of age). (**b**) Body weight evaluation of wild type (WT), FFAR3 knockout (FFAR3^−/−^), Tg2576 (Tg) and Tg2576 FFAR3 knockout mice (Tg-FFAR3^−/−^) subjected to HFD. (**c**) Glucose tolerance test (GTT) and (**d**) the corresponding AUC quantification of mice subjected to HFD. (**e**) Insulin tolerance test (ITT) and (**f**) the corresponding AUC quantification of mice fed on HFD. Results are expressed as mean ± SEM and circles in the subfigures represent each individual value (*n* = 6–8 in each group). * *p* < 0.05, ** *p* < 0.01, *** *p* < 0.001, **** *p* < 0.0001, two-way ANOVA followed by Tukey.

**Figure 3 ijms-23-13591-f003:**
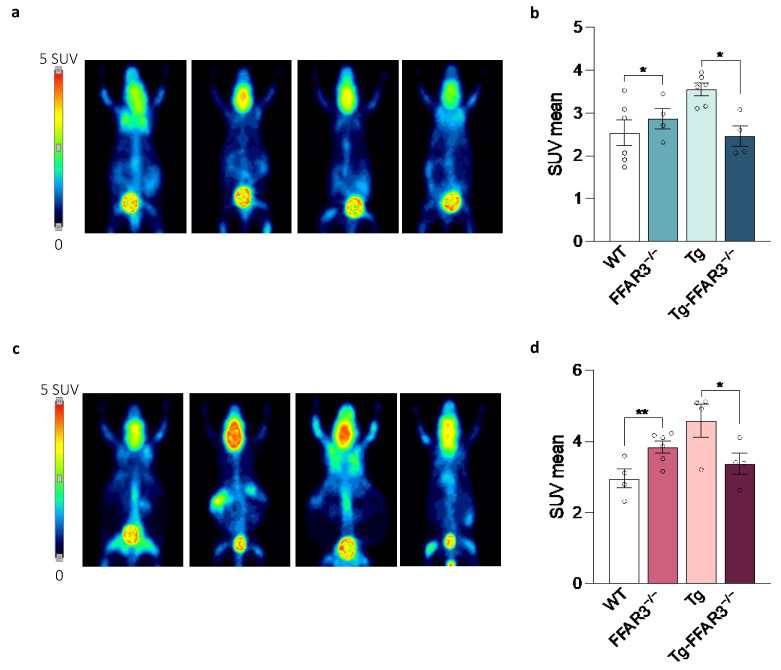
Tg2576 mice exhibit a brain glucose hypermetabolism which is restored upon FFAR3 deletion. (**a**) PET images of representative mice showing brain ^18^F-FDG signal in standard uptake value (SUV) and (**b**) the quantification of positron emission (SUV mean) of wild type (WT), FFAR3 knockout (FFAR3^−/−^), Tg2576 (Tg) and Tg2576 FFAR3 knockout mice (Tg-FFAR3^−/−^) fed with NCD. (**c**) PET images of representative mice showing brain ^18^F-FDG signal and (**d**) the quantification of positron emission (SUV mean) of mice subjected to HFD. Results are expressed as mean ± SEM and circles in the subfigures represent each individual value (*n* = 4–6 in each group). * *p* < 0.05 and ** *p* < 0.01, two-way ANOVA followed by Tukey.

**Figure 4 ijms-23-13591-f004:**
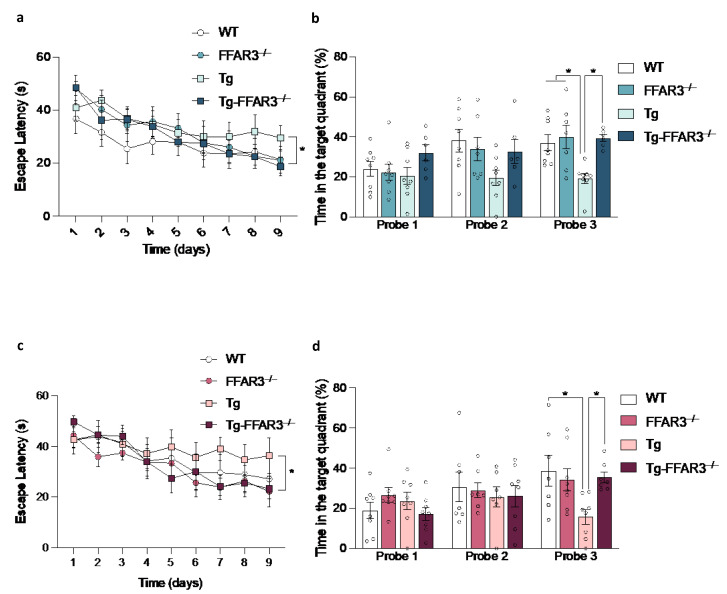
FFAR3 ablation prevents the cognitive deficiency observed in 15-month-old Tg2576 mice. Spatial memory evaluation by Morris water maze (MWM) (**a**) memory acquisition and (**b**) memory retention phase of wild type (WT), FFAR3 knockout (FFAR3^−/−^), Tg2576 (Tg) and Tg2576 FFAR3 knockout mice (Tg-FFAR3^−/−^) fed with NCD. MWM (**c**) memory acquisition and (**d**) memory retention phase of mice subjected to HFD. Results are expressed as mean ± SEM and circles in the subfigures represent each individual value (*n* = 6–9 in each group). * *p* < 0.05 two-way ANOVA followed by Tukey.

## Data Availability

The data that support the findings of this study are available from the corresponding author upon reasonable request.

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
