# Peer review of "The Cognitive Improvement and Alleviation of Brain Hypermetabolism Caused by FFAR3 Ablation in Tg2576 Mice Is Persistent under Diet-Induced Obesity"

_ijms, 2022, doi:10.3390/ijms232113591_

Round 1

Reviewer 1 Report

This study aims at ablation of the FFAR3 gene in Tg2576 mice and showed an cognitive improvement.

However there are some main questions remains unclear here:

1) What is the effect of ablation of the FFAR3 gene on tau and amyloid plaques?

2) What is the effect of ablation of the FFAR3 gene on connection between neurons?

Author Response

  • What is the effect of ablation of the FFAR3 gene on tau and amyloid plaques?

We thank the reviewer to point this issue to our attention. The impact of FFAR3 ablation on amyloid plaques and Tau phosporylation has been more deeply explained in the Discussion section (please see lines 466-478). Indeed, this idea has been extensively addressed in a previous paper published by our group in this Journal (Int J Mol Sci. 2022; 23(7):3533. doi: 10.3390/ijms23073533).

In that paper, the effect of FFAR3 knockdown on Aβ production and fibrillar Aβ depositions was determined. ELISA measurement in cortical homogenates showed barely detectable levels of Aβ42 in 12-month-old Tg-FFA3R−/− (named in the paper as FFAR3-/-/APPswe) mice, as opposed to what happened in age-matched Tg2576 mice (named in the paper as APPswe), which had high levels of this polypeptide.

Moreover, the APP-derived carboxy-terminal fragments (CTFs) processing in these animals by Western blot was analyzed, finding that there were no significant differences in the levels of the CTFs (C83 and C99) between FFA3R−/− vs. WT and Tg-FFA3R−/− vs. Tg2576 mice.

Interestingly, immunohistochemical analysis of hippocampal and frontal cortex sections using the anti-Aβ 6E10 antibody demonstrated that Tg-FFA3R−/− mice were completely free of Aβ deposits, which were abundant in the brain of the Tg2576 mouse.

Concerning this, it was found that the levels of insulin degrading enzyme (IDE), which reduces the accumulation of APP-derived toxic peptides, were increased in the hippocampus of FFA3R−/− and Tg-FFA3R−/− mice (FFA3R−/− vs. WT, p < 0.01; Tg-FFA3R−/− vs. WT, p < 0.001; Tg-FFA3R−/− vs. Tg2576, p < 0.01). Therefore, the Aβ reduction, both regarding soluble and fibrillar Aβ, could be a consequence of higher clearance mediated by Aβ-degrading enzymes. Whereas IDE was increased, no significant inter-group differences in the expression of neprilysin transcripts were detected.

Regarding Tau pathology, the 12-month-old Tg-FFA3R−/− mice displayed significantly reduced levels of phosphorylated tau, at the epitopes recognized by the AT8 antibody (Ser202/Thr205), compared with the Tg2576 transgenic mice.

When looking for changes in kinase activity accounting for the reduction in tau phosphorylation, an increase in the tau kinase GSK3β active form, phosphorylated at Tyr216, in Tg2576 mice compared with WT mice (p < 0.001) was observed. Tg-FFA3R−/− mice showed significantly lower levels of active GSK3β as compared with Tg2576 mice (p < 0.001) and similar to WT mice. In addition, the inactive form of GSK3β, phosphorylated at Ser9, was increased in Tg-FFA3R−/− vs. Tg2576 (p < 0.001).

It has been demonstrated that the kinase Cdk5, which also phosphorylates tau, is activated by p35 protein. Calpain-mediated cleavage of p35 to p25 (a 208-residue carboxy-terminal fragment of p35) p25 causes prolonged activation and mislocalization of Cdk5 and hyperphosphorylation of substrates such as tau. The analysis of the p25/p35 ratio in Tg2576 mice showed a significant increase (1.63 ± 0.07 (p < 0.01), whereas Tg-FFAR3 −/− (0.75 ± 0.10) remained with similar ratios to FFA3R−/− and WT.

2) What is the effect of ablation of the FFAR3 gene on connection between neurons?

This issue has also been extensively studied in the paper performed by Zamarbide et al. (Int J Mol Sci. 2022; 23(7):3533. doi: 10.3390/ijms23073533) and has been more deeply explained in the Discussion section (please see lines 504-509).

In order to address the connection between neurons different synaptic plasticity markers were assessed in the frontal cortex. First, the ionotropic AMPA receptors were found increased in both FFA3R−/− and Tg-FFA3R−/− mice. The results show a statistically significant increase in the phosphorylation of the AMPA subunit GluR1 in FFA3R−/− and Tg-FFA3R−/− mice compared with WT and Tg2576 mice, respectively (p < 0.001). Meanwhile, the analysis of GluR2 and GluR3 AMPA subunits exhibited an increase in their expression levels in Tg-FFA3R−/− vs Tg2576 mice and WT mice (p < 0.05).

Moreover, cytoskeletal protein PSD95, a member of the post-synaptic density, is decreased in Tg2576 mice (p < 0.05 vs. WT mice), and its levels are recovered in the Tg-FFA3R−/− mice (p < 0.001 vs. Tg2576 mice) and in FFA3R−/− compared with WT mice (p < 0.001). MAP-2 shows similar cortical levels in the WT, FFA3R−/−, and Tg-FFA3R−/− mice. Conversely, Tg2576 mice have decreased levels of MAP-2 compared with WT mice (p < 0.05).

Finally, the levels of the phosphorylated form of CaMK II are preserved in FFA3R−/− and Tg-FFA3R−/− mice, whereas they are decreased in the Tg2576 mice (p < 0.01 vs. WT mice). Comparing the transgenic animals, the expression was higher in Tg-FFA3R−/− than in Tg2576 mice.

Reviewer 2 Report

The manuscript presented for review takes up an interesting topic about the potential influence of genetic predisposition and the importance of receptors for medium chain fatty acids in the development of Alzheimer's disease. The authors of the manuscript clearly presented the possible mechanisms linking carbohydrate abnormalities and the risk of changes leading to the development of Alzheimer's disease. The obtained results indicate the importance of FFAR3 receptors for brain hypermtabolism and cognitive dysfunction in older animals, especially in the case of increased dietary fat intake. However, in this context, it would be worth presenting in the methodology of work information on the characteristics of fat (fatty acids) used in a high-fat diet. In addition, did the mice that increased their body weight during the intervention consumed significantly greater amounts of feed compared to the animals of the control group?

For a better look at the presented results, it is worth placing the chapter: Materials and methods traditionally, after the chapter: Introduction.

In the description of the statistical analysis, it is worth mentioning the computer program that was used for their analysis and including the figures with information on statistical differences (using tests for dependent variables).

Author Response

The manuscript presented for review takes up an interesting topic about the potential influence of genetic predisposition and the importance of receptors for medium chain fatty acids in the development of Alzheimer's disease. The authors of the manuscript clearly presented the possible mechanisms linking carbohydrate abnormalities and the risk of changes leading to the development of Alzheimer's disease. The obtained results indicate the importance of FFAR3 receptors for brain hypermetabolism and cognitive dysfunction in older animals, especially in the case of increased dietary fat intake.

However, in this context, it would be worth presenting in the methodology of work information on the characteristics of fat (fatty acids) used in a high-fat diet.

We thank the reviewer to point this issue to our attention. Indeed, the information about the employed diet was mistaken in the manuscript. The diet used in the study contained 35% calories from carbohydrates, 20% from protein and 45% from fat (Research Diets, #D12451). The fat content comes from lard (that contains 0.72mg/gr of cholesterol) and soybean oil (which typically features a fatty acid composition of 51% linoleic acid, 7-10% α-linolenic acid, 23% oleic acid, 10% palmitic acid and 4% stearic acid). This information has been corrected and completed in the material and method section (please see lines 525-526).

In addition, did the mice that increased their body weight during the intervention consumed significantly greater amounts of feed compared to the animals of the control group?

The reviewer has highlighted a very interesting idea. When comparing the effect of the diet in body weight gain, two-way ANOVA analysis showed a significant effect of the diet (F1, 54= 48.44, p<0.001), indicating that all the groups increased significantly their body weight due to the high fat feeding.

For a better look at the presented results, it is worth placing the chapter: Materials and methods traditionally, after the chapter: Introduction.

According to the International Journal of Molecular Sciences style, Material and Methods section must be placed at the end of the manuscript.

In the description of the statistical analysis, it is worth mentioning the computer program that was used for their analysis and including the figures with information on statistical differences (using tests for dependent variables).

The computer program used for the statistical analysis is Graph Pad Prism 9.0 (this is stated at the material and methods section, please see line 599). Moreover, all the information of the statistical significance (F and p values obtained in the analysis) is specified across the whole results section in brackets and straight after each presented result. For example: Body weight assessment indicated that Tg2576 mice feature significantly lower body weight (two-way ANOVA, main effect of Tg, F1,27= 10.83, p<0.01). (For more examples please see: lines 123, 125, 135-136, 140-141, 265-266 or 269-270).
